# Environmental Factors as Modulators of the Relationship between Obstructive Sleep Apnea and Lesions in the Circulatory System

**DOI:** 10.3390/jcm9030836

**Published:** 2020-03-19

**Authors:** Dominika Urbanik, Helena Martynowicz, Grzegorz Mazur, Rafał Poręba, Paweł Gać

**Affiliations:** 1Department of Internal Medicine, Occupational Diseases, Hypertension and Clinical Oncology, Wroclaw Medical University, Borowska 213, PL 50-556 Wroclaw, Poland; dominika.urbanik@student.umed.wroc.pl (D.U.); helena.martynowicz@umed.wroc.pl (H.M.); grzegorz.mazur@umed.wroc.pl (G.M.); rafal.poreba@umed.wroc.pl (R.P.); 2Department of Hygiene, Wroclaw Medical University, Mikulicza-Radeckiego 7, PL 50-368 Wroclaw, Poland

**Keywords:** obstructive sleep apnea, environmental factors, cardiovascular system

## Abstract

Obstructive sleep apnea (OSA) is a growing social problem, particularly in well-developed countries. It has been demonstrated that obstructive sleep apnea is a significant risk factor for cardiovascular diseases, including arterial hypertension, ischemic heart disease, heart failure, rhythm/conduction disturbances, as well as cerebral stroke. The pathophysiology of these diseases is complex and multifactorial. We present the current state of research on behavioral and environmental factors that influence the relationship between OSA and cardiovascular changes. We discuss the relationship between obesity, alcohol, sedatives, environmental tobacco smoke, allergic diseases and environmental pollution on the one hand and OSA on the other. In this context, the environment should be considered as an important modulator of the relationship between OSA and cardiovascular diseases.

## 1. Introduction

Obstructive sleep apnea (OSA) is a growing medical problem with socioeconomic ramifications, particularly in well-developed countries. It may affect any sex or age group. It occurs in about 14% of men and 5% of women aged 30–70. The prevalence of this illness grows with age. In women, it often develops in the post-menopausal period. Moderate and severe OSA occurs about 3 times more often in men than in women, in the group of middle-aged patients, between 30 and 49 years of age [1]. Other reports show a much more frequent occurrence of this illness—23.4% in women and 49.7% in men [2]. OSA is diagnosed in up to 70% patients undergoing bariatric treatment [3]. In post-stroke patients, the prevalence of OSA is 72% [3,4]. A study performed on a Polish population—Prospective Urban Rural Epidemiology Study (PURE)—showed that more than half of the subjects were at moderate or high risk of developing obstructive sleep apnea, in 66.5% of men and 60.1% of women, respectively [5].

Obstructive sleep apnea is characterized by repeated episodes of collapse of the upper airways during sleep, with continued functioning of the respiratory muscles, leading to apnea and hypopnea. Increased resistance in the airways is the cause of awakenings related to increased respiratory effort, so-called respiratory effort-related arousals (RERA). The adverse respiratory events lead to decreases in saturation, activation of the sympathetic nervous system, increased arterial pressure, alternate bradycardia and tachycardia, as well as arousals and sleep fragmentation [6].

The main risk factors of obstructive sleep apnea are obesity and increased neck circumference. Other significant factors include anatomical defects of the craniofacial structures, as well as otorhinolaryngological diseases including swollen palatine tonsils, elongated soft palate, swollen uvula, turbinate hypertrophy, nasal septum deviation, retrognathism and micrognathism. In OSA aetiology, one should take note of the growing level of environmental pollution, as well as an increase in the incidence of allergy-based rhinitis. Improper lifestyle, including smoking and the consumption of alcohol and sedatives before sleep, also plays an important role in the development of this illness [7].

The dominant symptom of obstructive sleep apnea is snoring and interrupted breathing during sleep. The prevalence of snoring in OSA is 75%–90% [8,9]. Nocturnal symptoms also include restless sleep, frequent arousals, excessive perspiration, motor agitation and nocturia. Nocturia occurs in 2% of mild OSA and 82% in severe OSA. Daytime symptoms are mostly excessive drowsiness and fatigue. Moreover, the patients report morning headaches (12%–18%), dryness of the oral mucous membrane, impairment of cognitive functions, problems with memory and concentration, mood deterioration, and a tendency towards depressive reactions and potency disorders [10].

The diagnostic gold standard for OSA is polysomnography, used to assess the respiratory distress index (RDI), consisting in the number of apneas, hypopneas and respiratory effort-related arousals (RERA) per one hour of sleep. Due to the low availability of oesophageal pressure sensors, which are used to measure the respiratory effort in RERA, the apnea–hypopnea index (AHI) is used in clinical practice. According to the guidelines of the American Academy of Sleep Medicine, obstructive sleep apnea is diagnosed at AHI ≥ 15/h in asymptomatic patients, and at AHI ≥ 5/h with accompanying clinical symptoms. There are three stages of OSA severity: mild OSA (AHI ≥ 5 and < 15/h), moderate OSA (AHI ≥ 15 and ≤ 30/h), as well as severe OSA (AHI > 30/h) [11].

## 2. OSA and Lesions in the Circulatory System

It has been demonstrated that obstructive sleep apnea is a significant risk factor for cardiovascular diseases, including arterial hypertension, ischemic heart disease, heart failure, rhythm/conduction disturbances, as well as cerebral stroke [12,13,14,15]. The pathophysiology of these diseases is complex and multifactorial. The numerous arousals disturb sleep architecture and reduce the length of slow-wave sleep and REM, which increases the activity of the sympathetic nervous system and reduces the activity of the parasympathetic nervous system. Moreover, alternating hypoxemia and reoxygenation promote the phenomenon of oxidative stress and inflammation. Chronic, interrupted hypoxemia contributes to activation of the proinflammatory nuclear factor NF-kappa B. Stimulation of the systemic inflammatory pathways favors endothelial dysfunction, increased blood coagulability, insulin resistance, development of atherosclerosis, as well as stimulation of the renin–angiotensin–aldosterone system [16]. Fluctuating changes in thoracic pressure cause expansion of the atrial walls, affecting stimulation of the mechanoreceptors, activation of ionic channels, and, most likely, development of atrial rhythm disorders, particularly atrial fibrillation. Atrial expansion also causes secretion of the atrial natriuretic peptide, causing nocturia, especially in severe OSA [17].

Cardiovascular diseases are the most common cause of death in the world. According to statistics from the World Health Organization, in 2016, 18 million people died of cardiovascular diseases, which constituted 31% of all deaths. A distinct majority of 85% was caused by myocardial infarction or cerebral stroke. Therefore, proper prophylaxis and treatment of cardiovascular diseases are crucial. An overview of the studies performed to date shows a relationship between untreated, severe apnea and an increase in general mortality, as well as that caused by cardiovascular issues [12].

### 2.1. OSA and Arterial Hypertension

Obstructive sleep apnea is an independent risk factor for arterial hypertension. The pathogenesis of hypertension in OSA is complex. It was proposed that intermittent hypoxemia, activation of the renin–andiotensin–aldosteron (RAA) system, endothelial dysfunction, sleep fragmentation and nocturnal fluid shift may play a role in increases in blood pressure. OSA episodes may up-regulate sympathetic activation, which acts on the chemoreflex and may consequently result in hypertension. In the Wisconsin Sleep Cohort Study, it was demonstrated that the probability of developing arterial hypertension during a four-year observation correlated with the severity of obstructive sleep apnea [15]. Similar results were obtained by Marin et al., who also found a lower risk of arterial hypertension in patients subject to treatment with constant, positive airway pressure (CPAP) [18]. The average pressure decrease during treatment with a CPAP device was 2.6 ± 0.5 mmHg and 2.0 ± 0.4 mmHg for systolic and diastolic pressure, respectively [19]. A greater pressure reduction, by 4.78 mmHg for systolic pressure and 2.95 mmHg for diastolic pressure, is observed in patients with resistant arterial hypertension [20]. The hypotensive effects of using a CPAP device are also largely dependent on the average duration of use, the original RDI, as well as on the type of daily variability of arterial hypertension (dipper vs. non-dipper). In general, patients who use a CPAP device for at least 4 h a day receive the greatest benefits. In a meta-analysis conducted on 1906 subjects, Fava et al. demonstrated that a higher original AHI correlates with a greater reduction of arterial pressure during CPAP treatment [21]. In the case of patients with non-dipper arterial hypertension treated with a CPAP device, we can expect better hypotensive results, compared to patients with dipper arterial hypertension [22].

### 2.2. OSA and Ischemic Heart Disease

Obstructive sleep apnea affects the development of ischemic heart disease by promoting endothelial dysfunction, atherosclerosis and metabolic disorders. The prevalence of OSA in patients with cardiovascular diseases is 46% [1], while in patients undergoing percutaneous coronary interventions (PCI) or coronary artery bypass graft, the prevalence of moderate-to-severe OSA has been reported to be as high as 63.7% [23]. The prevalence of moderate and severe OSA among patients subject to coronary angioplasty is high, amounting to 45.3%. A three-year observation of patients with ischemic heart disease demonstrated that the risk of acute coronary syndrome was higher with concurrent obstructive sleep apnea, amounting to 18.9%, compared to a 14% risk in patients without OSA [24]. Khan et al. conducted a meta-analysis on 4268 patients with moderate and severe OSA concerning the effectiveness of CPAP treatment in the prevention of cardiovascular episodes. They demonstrated a significant risk reduction by 57% when using a CPAP device for more than 4 h a day [25]. Different results were obtained in the Sleep Apnea Cardiovascular Endpoints (SAVE) study, which failed to demonstrate a significant reduction in the primary end point (a composite of cardiovascular death, myocardial infarction, stroke, hospitalization for unstable angina, heart failure or transient ischemic attack) among patients treated with CPAP. However, there was a trend toward a lower risk of stroke for those with good CPAP treatment adherence, as well as a lower risk of the composite end point of cerebral events. Among secondary end points, CPAP treatment was associated with significant reductions in daytime sleepiness and snoring, improvements in health-related quality of life and mood, and fewer days off work because of poor health. Yet, it should be noted that the average duration of using a CPAP device during sleep was only 3.3 h, and that the study excluded patients with severe daytime sleepiness with Epworth Sleepiness Scale (ESS) > 15 and with severe night-time hypoxemia with saturation < 80% for > 10% of the study [26]. Therefore, the results of the SAVE study should be interpreted carefully and emphasize the need for further studies.

### 2.3. OSA and Heart Failure

Patients with heart failure (HF) were observed to have a higher prevalence of sleep-disordered breathing (SDB), both in the form of obstructive and central sleep apnea, compared to the general population. Arzt et al. examined 6876 patients with symptomatic stable heart failure in New York Heart Association (NYHA) class ≥ II and left ventricular ejection fraction ≤ 45% against sleep-disordered breathing. The frequency of moderate and severe forms of SDB was 49% in men and 36% in women [27]. The coexistence of SDB and heart failure involves a significantly higher mortality rate. During about 7 years of observing patients affected with SDB and HF, a statistically significant correlation between night-time hypoxemia and death for any reason was observed. Every hour of sleep with recorded saturation < 90% increases the risk of death by 16.1% [28].

### 2.4. OSA and Rhythm and Conduction Disorders

Obstructive sleep apnea affects the occurrence of cardiac rhythm and conduction disorders by alternate hypoxia and reoxygenation, increased activity of the sympathetic nervous system, as well as by creating a negative thoracic pressure. Moreover, OSA may lead to cardiac remodeling and increased susceptibility to arrhythmia. Selim et al. demonstrated a double risk of any cardiac disorder during sleep in patients with moderate and severe apnea [29]. Atrial fibrillation (AF) and ventricular rhythm disorders are the most commonly reported arrhythmias in patients with obstructive sleep apnea. In the Sleep Heart Health Study, patients with severe OSA were more likely to suffer from atrial fibrillation, nonsustained ventricular tachycardia and complex ventricular ectopic beats, by four, three and two times, respectively [14]. Night-time saturation decreases related to obstructive sleep apnea were found to be an independent predictive factor of atrial fibrillations in patients < 65 years of age during a five-year follow-up [30]. Numerous studies demonstrated a relationship between OSA and increased risk of recurring atrial fibrillation after electrical cardioversion and percutaneous ablation, by as much as 25% [31]. Constant positive airway pressure therapy prolongs the time without recurrence of arrhythmia [32,33]. OSA is also known to have a connection with episodes of bradycardia, atrioventricular block pauses, atrial fibrillation and complex ventricular ectopy [34]. Most bradycardia and conduction disorder events occurred in patients with severe OSA, prolonged by periods of desaturation at night and low partial oxygen pressure in the morning, after waking up [35]. There is evidence associating obstructive sleep apnea with a predisposition towards occurrence of sudden cardiac death (SCD), particularly at night-time [36]. Gami et al. demonstrated in over 10,000 patients that OSA-related hypoxemia is a significant predictive factor for SCD, independent of other risk factors, such as age, arterial hypertension, ischemic heart disease, heart failure and ectopic ventricular beats. The strongest SCD predictors included average night-time saturation < 78% and lowest saturation < 78%, alongside age > 60 years and AHI > 20 [37]. A potential mechanism connecting obstructive sleep apnea with the occurrence of sudden cardiac death is an increase in QT dispersion, which favors severe ventricular rhythm disorders [38].

## 3. Environmental Factors and OSA

### 3.1. Obesity and OSA

Contemporary literature reports that obstructive sleep apnea is a new risk factor that modulates the cardiometabolic consequences of obesity. Obesity is the main cause of OSA. Obesity-related hemodynamic changes include an increase in the absolute volume of circulating blood, as well as an increase in the cardiac ejection and minute volume, effected by an increase in the cardiac contraction force and heart rate [39]. Thus, hemodynamic changes coexist with OSA. Chronic inflammation is also important in OSA pathogenesis, which concurrently with visceral obesity, hyperlipidemia and insulin resistance favors the formation of atherosclerotic plaque [40]. In turn, hyperkinetic circulation and metabolic disorders are the cause of arterial hypertension. These disorders cause complications in the form of a characteristic cardiac remodeling process: left ventricular hypertrophy, left atrial enlargement and deterioration of cardiac diastolic function [41]. The obese typically suffer from heart failure with a preserved ejection fraction [42]. Obesity and its consequences predispose people to coronary disease and cardiac rhythm disorders, particularly atrial fibrillation [30].

The prevalence of obstructive sleep apnea was about 45% in a population of obese adults, and a 10% weight gain predicted an approximately 32% increase in the AHI [43]. Numerous studies demonstrated a significant correlation between reduction of body weight and a decrease in the respiratory disturbance index. The relationship between the AHI decrease and body weight reduction is non-linear. The original AHI value and the degree of body weight reduction are significant predictive factors. It is estimated that for each kilogram lost, the AHI decreases by 0.6–1.0/h, which may give a chance for a complete recovery from OSA if the illness is mild. The greatest benefits were observed in men with severe apnea who lost at least 10 kg [44]. Peppard et al. demonstrated that a 10% body weight reduction decreases the AHI by 26% [43]. There are reports that CPAP treatment supports the weight loss process. When comparing groups treated with the same diet and additionally treated with CPAP or intraoral devices, it was noted that only the group of patients supported with a CPAP device was able to significantly reduce their body weight [45]. Modification of environmental factors, namely, overweight and obesity, is the foundation of treating obstructive sleep apnea, along with its cardiological consequences. Only mild forms of OSA can be treated with diet alone. Complete treatment of severe OSA is achievable by employing bariatric surgery methods. In a meta-analysis conducted on 22,000 patients, bariatric surgeries involved an 85.7% treatment rate of obstructive sleep apnea [46].

### 3.2. Alcohol, Sedatives and OSA

The impact of alcohol and sedatives on triggering or increasing obstructive sleep apnea has been known for years [47,48]. Alcohol and sedatives increase the AHI by reducing tension of the muscles that keep the upper airways passable and by reducing the respiratory drive [49,50]. Arousals related to respiratory episodes restore the tension of the pharyngeal muscles and open blocked airways. It is a specific mechanism of defence against desaturation in patients with obstructive sleep apnea. Alcohol and sedatives increase the arousal threshold, which delays the opening of the airways and intensifies hypoxia and hypercapnia [51]. A meta-analysis of studies between 1985 and 2015 demonstrated that consumption of alcohol before sleep increases the risk of OSA by 25% [52]. Use of benzodiazepines by patients with obstructive sleep apnea involves a higher risk of acute respiratory failure [53]. The basis for treating OSA and its cardiological consequences is modification of environmental factors that intensify apnea, including the avoidance of alcohol and sedatives before sleep.

### 3.3. Tobacco Smoking

Another environmental factor that modulates the intensity of obstructive sleep apnea and its effects on the circulatory system is tobacco smoking. Active smokers are 2.5 times more likely to develop OSA than former and non-smokers [54]. Studies indicate that the OSA risk factor is current exposure, rather than past exposure to tobacco smoke [55]. The relationship between tobacco smoking and OSA can be explained by two mechanisms. First, smoking causes a chronic inflammation of the nasopharynx, edema of the mucous membrane, as well as increased secretion of mucus [56]. Second, a reduced nicotine concentration at night disturbs the neuromuscular reflexes and, consequently, reduces muscular tension, which may cause collapsing of the upper airways [57]. A study performed on over 5000 patients demonstrated that cigarette smoking correlates with a more severe form of obstructive sleep apnea and greater daytime sleepiness [58]. Similar results were obtained in a study by Hsu et al., where it was additionally found that smokers with OSA have a shorter total sleep time, reduced deep sleep and longer snoring period [59]. One should also note the synergic, adverse effect of obstructive sleep apnea and tobacco smoking on the circulatory system. Cigarette-smoking patients with severe OSA have a higher cardiovascular risk than smokers with mild and moderate apnea and non-smokers [60]. A study carried out by Levie designated biochemical cardiovascular risk factors, as well as markers of inflammation and oxidation stress, in smoking and non-smoking patients with obstructive sleep apnea. Cigarette smoking involved much higher levels of C-reactive protein, ceruloplasmin, haptoglobin and triglycerides, as well as a lower concentration of high-density lipoprotein (HDL) cholesterol. Smokers with severe OSA had the highest level of ceruloplasmin and the lowest level of HDL [60]. The disadvantageous distribution of inflammation markers and lipoprotein profile in smokers with OSA indicates a high oxidation potential, which increases the risk of atherosclerosis and cardiovascular diseases. Therefore, medical activities and social attitudes that support quitting tobacco are crucial.

### 3.4. Allergic Diseases, Environmental Pollution and OSA

The important environmental factors that affect OSA and its cardiological consequences are allergic diseases and environmental pollution. Climate changes, which have intensified in the last few decades, increased CO_2_ emissions and air pollution are closely related to the growing incidence of allergic respiratory diseases. Excessive exposure to particulate matter (PM) intensifies the symptoms of asthma and allergic rhinitis [61,62]. Xian et al. demonstrated that air-suspended, fine particulate matter of less than 2.5 µm (PM 2.5) damages the mucous membrane of the nasal cavity by loosening cellular connections and increases the release of proinflammatory cytokines [63]. This mechanism explains the excessive exposure to allergens and increased incidence of allergic rhinitis and asthma. Allergic rhinitis (AR) is one of the main reasons for nasal obstruction. It was demonstrated that patients with AR are 1.8 times more likely to experience moderate or severe obstructive sleep apnea than patients without allergies [64]. As AR is a modifiable OSA risk factor, it is necessary to implement appropriate diagnostics and treatment for patients with obstructive sleep apnea who report symptoms of chronic nasal obstruction. According to the available studies, nasal application of mometasone or oral application of montelukast reduces the intensity of mild obstructive sleep apnea in children [65,66]. Another way of alleviating the symptoms of nasal obstruction may be the use of household air filters [67]. However, so far, no studies analyzing the direct impact of air filters on obstructive sleep apnea have been conducted.

Increasing environmental pollution and climate changes are a global problem of the highest order that threatens the life and health of our planet. The observed global temperature increase is related to increased emissions of greenhouse gases and suspended particulate matter. Climate change is also affecting the intensification and consequences of obstructive sleep apnea. A study conducted by Weinreich et al. demonstrated that in all seasons, the increases in temperature and ozone concentrations at ground level resulted in about a 10% rise in OSA in the general population [68]. Similar results were obtained by Zanobetti et al., who additionally found a positive correlation between the number of respiratory episodes and the concentration of particulate matter under 10 µm (PM 10) [69]. Taking into account the profound and negative impact that obstructive sleep apnea has on the circulatory system, one should be aware that the fight against global warming is at the same time a fight for the health of our hearts.

## 4. Conclusions

In summary, the environment should be considered as an important modulator of the relationship between OSA and cardiovascular diseases. Behavioral and environmental factors (obesity, alcohol, sedatives, environmental tobacco smoke, allergic diseases and environmental pollution) significantly influence the relationship between OSA and cardiovascular changes (Figure 1).

Environmental and behavioral factors may impact the clinical presentation of OSA and the effectiveness of treatment. Inconsistent application of continuous positive airway pressure treatment by adults with obstructive sleep apnea is a common issue. Thus, future research is needed to explore the underlying mechanism of variable response to treatment, especially in different phenotypes of OSA exposed to various behavioral and environmental factors.

## Figures and Tables

**Figure 1 jcm-09-00836-f001:**
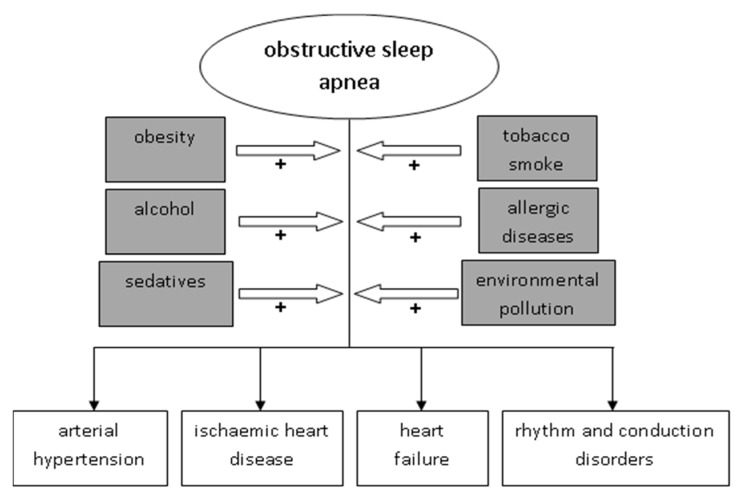
Environmental factors as modulators of the relationship between OSA and CVD. OSA, obstructive sleep apnea; CVD, cardiovascular diseases.

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
