# Peer review of "Environmental Factors as Modulators of the Relationship between Obstructive Sleep Apnea and Lesions in the Circulatory System"

_jcm, 2020, doi:10.3390/jcm9030836_

Round 1
Reviewer 1 Report
A very good review. Two minor errors
Page 2, line 56, The clinical practice admits the apnea hypopnea index (API). API is an incorrect abbreviation.
Page 5 line 191
Increases the risk of OBS by 25%. Do you mean OSA?
This review is a clear description of the various factors on sleep apnea. Providing tables or figures would make the review clearer. However there are a couple of typo errors that need to be corrected.
Obstructive sleep apnoea (OSA) is a growing social problem, particularly in well-
13 developed countries. It was demonstrated that the obstructive sleep apnoea is a significant risk factor
14 for cardiovascular diseases, including arterial hypertension, ischaemic heart disease, heart failure,
15 rhythm/conduction disturbances, as well as cerebral stroke. The pathophysiology of these diseases is
16 complex and multifactorial. We present the current state of research on behavioral and
17 environmental factors that influence on the relationship between OSA and cardiovascular changes.
18 We discuss relation between obesity, alcohol, sedatives, environmental tobacco smoke, allergic
19 diseases and environmental pollution on the one hand and OSA on the other. In their context, the
20 environment should be considered as an important modulator of OSA-cardiovascular diseases
21 relationship.
Author Response
Dear Reviewer,
Thank you for careful and thorough reading of this manuscript and for the thoughtful comments. We are very grateful for the review.
Comment of Reviewer: Providing tables or figures would make the review clearer.
Changes carried out in the paper: Figure 1 entitled "Environmental factors as a modulator of dependence between OSA and CVD" has been added.
Comment of Reviewer: Page 2, line 56, The clinical practice admits the apnea hypopnea index (API). API is an incorrect abbreviation.
Changes carried out in the paper: The API has been changed to AHI.
Comment of Reviewer: Page 5 line 191, Increases the risk of OBS by 25%. Do you mean OSA?
Changes carried out in the paper: OBS has been changed to OSA.
Best regards,
Authors

Reviewer 2 Report
This review comes across as somewhat superficial and unsophisticated. The writing is often difficult to follow. At times the authors cite review articles and not the primary literature despite this paper being a review article.
- Introduction
Line 25: Would not consider OSA as much a social problem as a medical problem with socioeconomic ramifications
31: The term “post-cerebral stroke” is confusing, particularly given that it has nothing to do with bariatric treatment. Reword for clarity?
Lines 48-53: Would be good to note what percent of patients report the mentioned symptoms. Your writing makes it sound like all patients have those symptoms.
Line 56-57: Do not know what these sentences mean. Clinical practice uses AHI instead of RDI? Would explain.
Line 56: I think you mean AHI not API
Line 60: OSA not ASO
Section 2.
Lines 63-65; 66-76: Would include citations.
73, 76: Would include references that are not review articles. One should generally avoid citing review articles in a review article.
Section 2.1.
Would consider expanding this section.
Section 2.2.
Lines 101-104: Would include citations.
Lines 115-16: Would soften language (given that this was the largest trial to address this question) and suggest what trial characteristics would help answer the question of cardiovascular disease prevention with CPAP.
Section 3.1.
Lines 156-7: Obesity is the main cause of the OSA, characterized by its negative impact on the circulatory system: This makes it sound like Obesity causes OSA by means of circulatory compromise. Is that what was intended?
Conclusion
Would include directions for future research.
Author Response
Dear Reviewer,
Thank you for careful and thorough reading of this manuscript and for the thoughtful comments. We are very grateful for the review.
Comment of Reviewer: Line 25: Would not consider OSA as much a social problem as a medical problem with socioeconomic ramifications.
Changes carried out in the paper: The “social problem “has been changed to “medical problem with socioeconomic ramifications”
Comment of Reviewer: Line 31: The term “post-cerebral stroke” is confusing, particularly given that it has nothing to do with bariatric treatment. Reword for clarity?
Changes carried out in the paper: The term “post cerebral stroke: has been changed to “patients after stroke”. This fragment has been changed to be more clear.
Comment of Reviewer: Lines 48-53: Would be good to note what percent of patients report the mentioned symptoms. Your writing makes it sound like all patients have those symptoms.
Changes carried out in the paper: The percent has been noted.
Comment of Reviewer: Line 56-57: Do not know what these sentences mean. Clinical practice uses AHI instead of RDI? Would explain.
Changes carried out in the paper: The differences between AHI and RDI has been explained.
Comment of Reviewer: Line 56: I think you mean AHI not API.
Changes carried out in the paper: The incorrect API has been changed to AHI.
Comment of Reviewer: Line 60: OSA not ASO.
Changes carried out in the paper: The mistake has been corrected.
Comment of Reviewer: Lines 63-65; 66-76: Would include citations.
Changes carried out in the paper: The citation has been included into text.
Comment of Reviewer: 73, 76: Would include references that are not review articles. One should generally avoid citing review articles in a review article.
Changes carried out in the paper: The references that are review has been removed and original paper has been cited.
Comment of Reviewer: Section 2.1.: Would consider expanding this section.
Changes carried out in the paper: The section has been expanded.
Comment of Reviewer: Lines 101-104: Would include citations.
Changes carried out in the paper: The citation has been included into text.
Comment of Reviewer: Lines 115-16: Would soften language (given that this was the largest trial to address this question) and suggest what trial characteristics would help answer the question of cardiovascular disease prevention with CPAP.
Changes carried out in the paper: The text has been changed and the language has been soften.
Comment of Reviewer: Lines 156-7: Obesity is the main cause of the OSA, characterized by its negative impact on the circulatory system: This makes it sound like Obesity causes OSA by means of circulatory compromise. Is that what was intended?
Changes carried out in the paper: The text has been changed to be more clear.
Comment of Reviewer: Would include directions for future research.
Changes carried out in the paper: The direction in future research has been indicated.
Best regards,
Authors

Round 2
Reviewer 2 Report
Thank you, you addressed each of my stated concerns.
This is a superficial, unsophisticated review of a complicated topic. The writing is quite poor and often difficult to follow. At times the authors cite review articles and not the primary literature despite this paper being a review article.
Statements like this will need to be heavily edited for clarity:
The future research are needed to improve the effectiveness of treatment in different phenotypes of OSA exposed to varius behavioral and enviromental factors.
Author Response
Dear Reviewer,
Thank you for careful and thorough reading of this manuscript and for the thoughtful comments. We are very grateful for the review.
Comment of Reviewer: The writing is quite poor and often difficult to follow.
Changes carried out in the paper: The whole text of the paper was checked by the native-speaker. The manuscript was corrected regarding English language syntax and grammar.
Comment of Reviewer: At times the authors cite review articles and not the primary literature despite this paper being a review article.
Changes carried out in the paper: All the reviews has been removed from literature. The only three (18,47,53) present also metaanalyzes, which present data relevant to the manuscript.
Comment of Reviewer: Statements like this will need to be heavily edited for clarity: "The future research are needed to improve the effectiveness of treatment in different phenotypes of OSA exposed to varius behavioral and environmental factors."
Changes carried out in the paper: The fragment has been changed to make it more clear: "The future research are needed to explore the underlying mechanism of variable response to treatment, especially in different phenotypes of OSA exposed to various behavioral and environmental factors."
Best regards,
Authors
